

# Adiposity and body fat distribution based on skinfold thicknesses and body circumferences in Czech preschool children, secular changes

Anna Vážná[1], Jan M. Novák[1], Robert Daniš[2] and Petr Sedlak[1,2]

[1] Department of Anthropology and Human Genetics, Charles University, Prague, Czech Republic
[2] Division of Child Health Promotion, Department of Hygiene, Third Faculty of Medicine, Charles University Prague, Prague, Czech Republic

Corresponding author
Anna Vážná, vaznaa@natur.cuni.cz

## ABSTRACT

**Background:** The long-standing widespread prevalence of obesity includes issues of its evaluation. Nutritional status may be assessed using various tools and methods; among others simple anthropometric measurements are well established. Widely used body mass index (BMI), presents an obstacle of needing to calculate a standard deviation score (SD) for correct use in the child population. As BMI overlooks body composition, it is necessary to evaluate fat and muscle mass with different methods. Established skinfolds and circumferences are used in many variations and equations to accomplish that goal; however, the parameters used in these methods also undergo secular changes. Furthermore, secular changes have been documented in fat mass distribution. The aim of the study is to assess secular changes of skinfolds thickness and body circumferences and evaluate their validity for use in clinical practice and population research.

**Methods and sample:** Our database consisted of a recent (2016–2022) sample with 594 participants (298 males) and a reference sample (from 1990) with 2,910 participants (1,207 males). Both cohorts comprised Czech preschool children, aged 4.00 to 6.99 years. With standard methodology, anthropometric parameters were obtained for 13 skinfolds and eight circumferences, by trained staff. The equations of Slaughter, Durnin and Deurenberg were correspondingly calculated. Statistical evaluation was conducted in the R programming language, using Welch's test, Cohen's d and the Bland–Altman method.

**Results:** Our study found significant increases in skinfold thickness on the abdomen, chest I. and forearm, with high clinical relevance ($p \geq 0.01$; $d$ = range from 0.20 to 0.70). Contrastingly, apart from the abdominal area, a decrease of circumferences was observed. The body fat percentage estimation equations were tested for bias in the recent sample in the context of bioimpedance analysis with the Bland–Altman method. All equations are suitable for application in clinical use.

**Discussion:** Documented secular changes in fat mass distribution are only part of a contemporary accelerating trend of obesity prevalence. Our findings support the trend of a decline of circumferences and rise of skinfold thickness in corresponding areas, especially on the limbs, that is evidenced by the trend of latent obesity. The results of the study show the need to complement established diagnostic procedures in childhood obesitology with abdominal and midthigh circumferences and

optionally even the maximal circumference of the forearm. These circumferences should always be measured alongside the skinfold thickness of the region. Only in this way can the overall adiposity of an individual with regard to secular changes, including the detection of latent obesity, be objectively evaluated.

## INTRODUCTION

Obesity is a well-known and long-standing problem in children as well as adults worldwide. The World Health Organization states that in 2022 there were over 390 million children and adolescents who were overweight, including within that number 160 million children living with obesity (*World Health Organisation, 2024*). With an increasing trend, the pandemic of children who are overweight and obese has been an alarming problem with an aggravating tendency heightened by the recent COVID-19 pandemic (*Pietrobelli et al., 2020*; *Stavridou et al., 2021*; *Wu et al., 2022*; *Vážná et al., 2022*). This is a global challenge that highlights the contemporary importance of evaluating children's nutritional status, which is traditionally assessed using body mass index (BMI) as a baseline. However, this approach is often inaccurate, as the proper evaluation for children requires calculating the BMI standard deviation score (BMI-SDS), which efficiently accomodates for age-related changes and differences between sexes. BMI must therefore be used in relation to sex and age reference values according to the WHO (*de Onis et al., 2004*; *de Onis, 2007*) or national reference standards (*Vignerová et al., 2006*). However, BMI-SDS as screening tool does not reflect body composition or the amount of fat mass, which is the main health concern and needs to be assessed albeit that BMI-SDS and %BF (percentage of body fat) are reportedly highly correlated (*Katzmarzyk et al., 2015*; *Vanderwall et al., 2017*). The need for evaluation of weight status relates to the comorbidities which excess weight and especially high %BF bring. These health concerns vary from cardiovascular diseases (*de Kroon et al., 2010*; *Chung, Onuzuruike & Magge, 2018*), to dyslipidaemia (*Burlutskaya et al., 2021*), prediabetes and diabetes mellitus type 2 (*Wabitsch et al., 2004*; *Ek et al., 2015*), tumors (*Aarestrup et al., 2020*) and other conditions, alongside the lower lifespan that accompanies children with obesity turning into adults with obesity (*Etzel et al., 2022*). Almost three quarters of adults with obesity did not suffer obesity in childhood, but in just over half of children with obesity continues with the condition into adolescence with the persisting condition, and 80% of adolescents with obesity continue to have obesity in adulthood (*Simmonds et al., 2015*).

The long-term rise in prevalence suggests a need for a reliable and valid tool for weight status evaluation. Using only BMI-SDS might raise obstacles as the index does not reflect body composition (*Etchison et al., 2011*). In the diagnosis of obesity it is necessary to prove an increase of body fat mass (*de Onis, 2007*), or an increase in the ratio of fat mass to muscle mass (*Sedlak et al., 2020*). Information regarding fat mass can be acquired using anthropometry, bioelectrical impedance analysis (BIA) and more sophisticated medical

devices. The need to take body composition into consideration is illustrated by a study of German children from 2005/2006; when divided into groups (underweight, normal-weight and overweight) according to BMI-SDS and compared to a reference sample from 1975, the phenomenon of body fatness measure was found to increase with skinfold caliper and was noted only in the underweight and normal weight groups. The study showed no significant differences between the recent group and the 1975 group in the overweight category (*Kromeyer-Hauschild, 2012*). Therefore, the need for assessment of body composition is unavoidable.

The traditional methods for evaluating body composition, *i.e.* anthropometric characteristics such as skinfold thickness and circumferences, are an established part of the evaluation of the nutritional status of children. Besides obesitology and endocrinology, child oncology also evaluates nutritional status with the use of selected markers, which are usually triceps and subscapular skinfolds, alongside with arm circumference (*Viani et al., 2020*; *Yaprak et al., 2021*; *Priyanka et al., 2022*). Likewise, the skinfold above the biceps and the suprailiac skinfold are also used (*Andaki et al., 2014*). Additionally, circumferences such as those of the waist, abdomen and hips are used for various evaluations (*Andaki et al., 2014*, *2018*). For clinical use and population studies, the subscapular and above triceps skinfolds are mostly used. This combination enables at least an indicative assessment of subcutaneous fat distribution in the trunk *vs.* limbs relationship, and therefore evaluate forms of distribution that are high-risk (*Marrodán Serrano et al., 2015*; *Ramírez-Vélez et al., 2016*; *Tang et al., 2020*). Circumferences are also used to define the level of metabolic risk; the most valid, for both children and adults, has proven to be the waist to height ratio (WHtR index) (*Savva et al., 2000*).

Another option for the estimation of adiposity in clinical practice and population studies is the use of simple regression equations such as *Slaughter et al. (1988)*, Durnin (*Durnin & Womersley, 1974*), Deurenberg (*Weststrate & Deurenberg, 1989*) and the sum of four skinfolds (*Vignerová & Bláha, 2001*). These equations use the regression relationship of selected skinfolds to estimate the percentage of subcutaneous fat as a correlate of overall adiposity. Each uses skinfolds and/or circumferences and has its own limitations. The first limitation is the population specificity of the equations, as use on different populations can influence and overestimate body fatness (*Dezenberg et al., 1999*).

Studies have proven a significant rise in adiposity, and therefore an increase in the prevalence of overweight and obesity in recent children across populations (*Liem et al., 2009*; *Nagel et al., 2009*; *Sedlak et al., 2015*; *Leal et al., 2015*; *Suder, Gomula & Koziel, 2017*). This is not the only secular trend as there is also the change and influence of the distribution of fat mass, which can alter due to lifestyle, even in child populations. With the rise in incidence of abdominal distribution in children (*Huang et al., 2001*; *Suder, Gomula & Koziel, 2017*; *Żegleń et al., 2022*), some methods can lose validity due to this secular change. This is supported even by the measurement of body fat by use of bioimpedance (BIA), which is in good agreement with the clinical gold standard, dual x-ray absorptiometry (DXA). In comparison to BIA, studies have reported that equations such as Slaughter and others have a tendency to underestimate body fatness (*Noradilah et al., 2016*; *Moeng-Mahlangu et al., 2022*). However, different methods must be used with caution, as their

interchangeability is limited (*Forte et al., 2021*). Nevertheless, secular changes are another challenge that needs to be taken into account as studies show substantial changes in child populations (*Sun et al., 2012*; *Suder, Gomula & Koziel, 2017*).

Owing to the documented secular changes in the form of the distribution of subcutaneous fat and thus the thickness of skinfolds and the sizes of body circumferences, the task of this study is to analyze the changes of selected skinfolds and circumferences in current Czech preschool children in relation to a reference cohort from 1990, and to evaluate their validity for use in clinical diagnostics and population research. The significance of our study lies in bringing attention to secular changes in markers used in the evaluation of adiposity. These findings are pivotal in practical use as well as to understanding the changes that are developing in contemporary child populations.

## MATERIALS AND METHODS

This cross-sectional study compared Czech preschool children from 1990 (the reference sample) to a recent sample (collected between 2016 and 2022). The study focused on children aged 4.00 to 6.99 years, with exclusions made for those outside this range (*n* = 32) and those with missing observations (*n* = 215), resulting in a total of 2,910 participants (594 in the recent and 2,316 in the reference samples). The 1990 reference sample was provided by *Bláha (1990)*. Children were measured from across the Czech Republic. This sample is one of last detailed national studies in the Czech Republic conducted in 1990. Data ascertained in this sample remain valid for the Czech preschool population as criteria of reference. For comparison with our recent sample, the original database was used. Children in the recent sample were measured in kindergartens in Prague and the surrounding area. In both samples recruitments were conducted *via* addressing kindergartens and schools. Upon agreement from the principal's cooperation in the study, legal guardians were informed and given information alongside informed consent to sign. The representativeness of both samples was ensured by the broad addressing of educational institutions and legal guardians of all children without selection. The response rate in the recent sample was 73%, the response rate for the reference sample is not available. All children lived in large towns and cities in comparable environmental conditions. Children were participating with the consent of their legal representatives.

For a detailed overview of sample and subsample sizes, please see Table 1.

Ethical approval was obtained from the Institutional Review Board of the Faculty of Science, Charles University, Prague (IRB approval number 2017/23), and the study was conducted in accordance with the Declaration of Helsinki (Fortaleza actualization). Written informed consent was obtained from parents, who agreed to anthropometric measurements being taken and the bioelectrical impedance analysis of their children in the 2016–2022 sample. All data was anonymized for confidentiality.

Anthropometric measurements were performed in accordance with standard techniques (*Martin & Saller, 1957*; *Eston et al., 2009*) by trained staff. Body height was assessed using a portable stadiometer (Trystom, s.r.o., Olomouc, Czech Republic), with an exactitude of 1 mm. Body weight was measured within the framework of BIA with an accuracy of 0.1 kg. BMI was calculated as weight (in kilograms) divided by square height

**Table 1 Study sample.**

| Age | Sex | Reference sample (1990) | | Recent sample (2016 to 2022) | |
|---|---|---|---|---|---|
| | | F | M | F | M |
| 4.00–4.99 | | 315 | 323 | 124 | 118 |
| 5.00–5.99 | | 390 | 364 | 94 | 94 |
| 6.00–6.99 | | 404 | 520 | 78 | 86 |

Note:
F, girls; M, boys.

(meters). BMI-SDS was then calculated using WHO standards. Categories were created with WHO standard guidelines from −2 to 1 SD as normal-weight, >1 SD overweight, >2 SD obesity (*de Onis et al., 2004*; *de Onis, 2007*). For comparison with the reference sample, 13 skinfolds were measured by the same technique, using a modified Best (Trysom) skinfold caliper with accuracy to 1 mm, calibrated to 2 N. This technique was used to measure skinfolds on the trunk (chest I, chest II, abdomen, suprailiac and subscapular), on the upper and lower extremities (on the biceps, triceps, forearm, mid-thigh, over patella, calf I, calf II) and above the hyoid bone. At same time four skinfolds (suprailiacal, subscapular, on the biceps and triceps) were measured with Harpenden calipers (with accuracy to 0.1 mm) for use in calculating body fat percentage.

The body circumferences were measured with a tape to the nearest 1 mm (chest on the mesosternale level, abdomen over the omphalion, gluteal as widest part over the hips, middle of relaxed arm, maximal of forearm, midpoint of the thigh, maximal of calf).

Body fat percentage estimation used the sum of four skinfolds method, a regression equation according to Slaughter (*Slaughter et al., 1988*), Durnin (*Durnin & Womersley, 1974*) and Deurenberg (*Weststrate & Deurenberg, 1989*).

Bioimpedance analysis was conducted using InBody 230 (DMS-BIA Technology; InBody Co., Seoul, Korea), which is declared by the manufacturer to be applicable for children from the age of 3 to estimate body composition.

Statistical analysis was conducted in the R programming language version 4.3.2 (R Core Team) and jamovi version 2.5 (*The jamovi project, 2024*). Groups were compared using Welch's test to mitigate potential biases stemming from unequal variances. Effect sizes were estimated using Cohen's d, with interpretations based on Cohen's original guidelines: small ($d = 0.2$), medium ($d = 0.5$), and large effects ($d = 0.8$) (*Cohen, 1988*). The Bonferroni correction was applied to reduce the risk of false significance resulting from multiple comparisons. The association between continuous parameters was assessed using basic linear models. To verify the validity of methods for determining body fat percentage, a simple correlation analysis was used alongside assessment of bias using the Bland–Altman method.

## RESULTS

First, the thicknesses of thirteen skinfolds of the recent sample of children were compared with the reference sample, see Figs. 1 and 2. In both sexes, the greatest thickness in both sets was the skinfold on the thigh, the lowest thickness for the skinfold on the chest II.

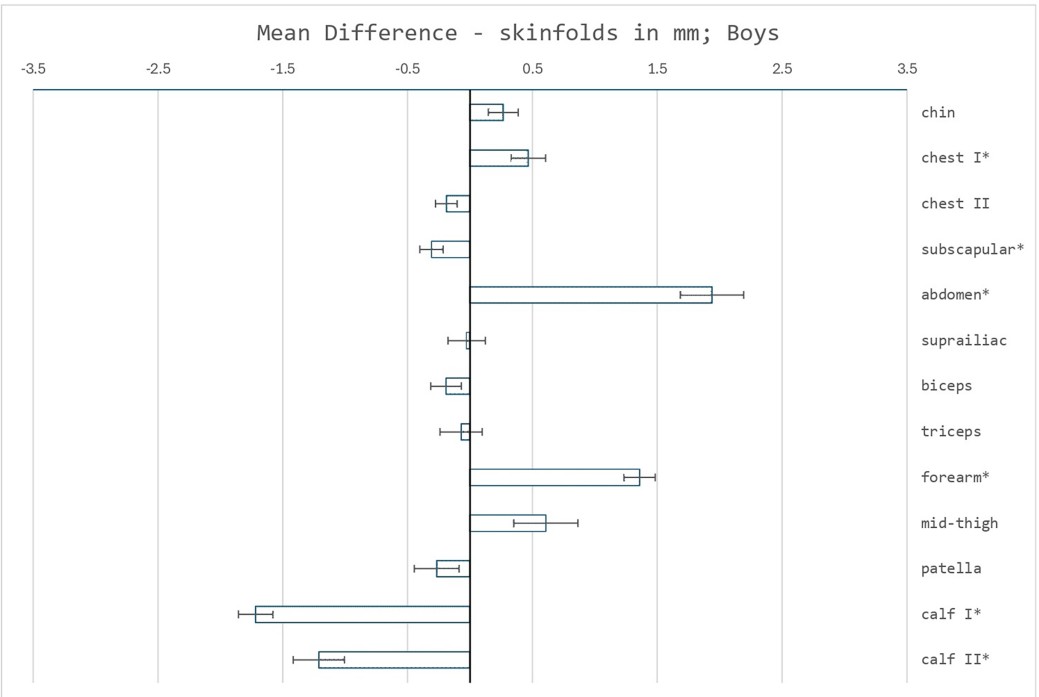

**Figure 1 Mean differences in skinfold measurements (mm) between the reference sample and the recent sample of boys.** Error bars represent the standard error of the mean difference. Asterisks (*) indicate significant differences determined using Welch's test, with $p < 0.05$ adjusted using the Bonferroni correction.

Significant increases were detected, without intersex differentiation, in the skinfolds on chest I, forearm and mainly the abdomen. High clinical relevance, effectivity of a marker in implication for population heath, was confirmed for all. In contrast, several skinfolds showed both statistically and clinically significant decreases—in girls, skinfolds on calf I and II, and also above the patella, in boys only the skinfolds on calf I and II. Significant differences were not confirmed in the diagnostically used skinfolds above the triceps, above the biceps, the subscapular and the suprailiacal.

Changes in circumferences in the areas of the measured skinfolds showed, apart from the circumference of the abdomen, a decreasing trend (see Figs. 3 and 4). A significant decline of all listed circumferences was detected in both sexes. The highest clinical relevance was in the circumferences of the forearm and calf. Conversely, a significant increase was found only in the abdominal circumference in girls, with high clinical relevance.

Evaluation of change in body-fat percentage between the recent and reference samples using the Slaughter equation showed a significant increase in girls ($p \geq 0.01$), with medium to high clinical relevance ($d = 0.78$). In boys, an increase in adiposity was supported by the sum of four skinfolds method ($p \geq 0.01$), but without clinical relevance ($d = 0.18$) (Tables 2 and 3). The BMI standard deviation value, meanwhile, remained stable in both sexes and showed no changes between samples.
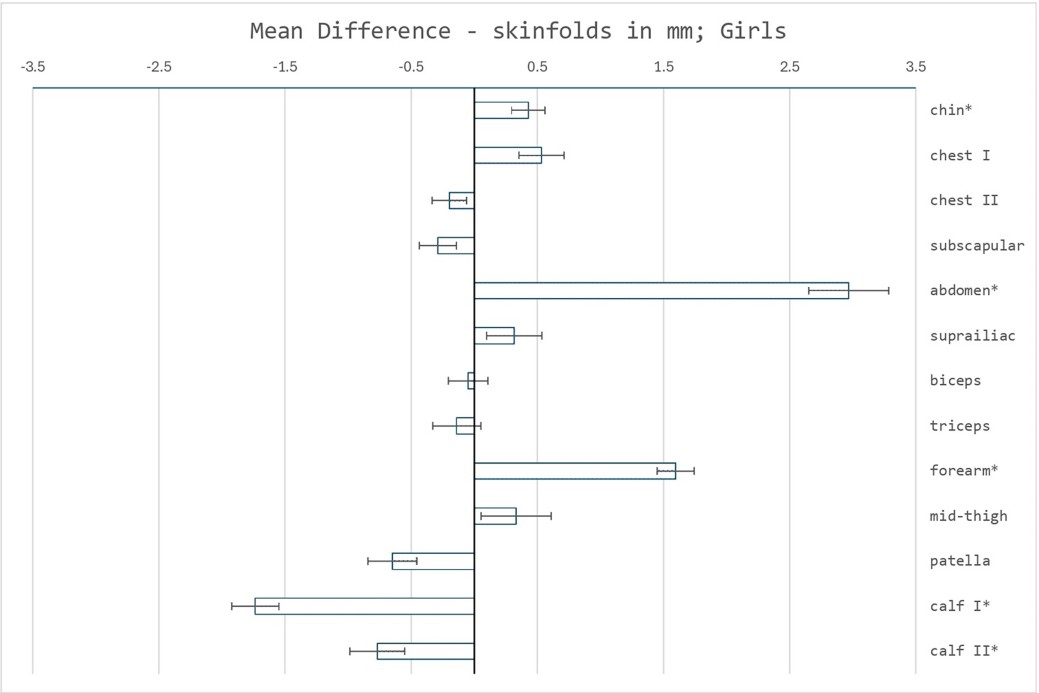

**Figure 2 Mean differences in skinfold measurements (mm) between the reference sample and the recent sample of girls.** Error bars represent the standard error of the mean difference. Asterisks (*) indicate significant differences determined using Welch's test, with $p < 0.05$ adjusted using the Bonferroni correction.

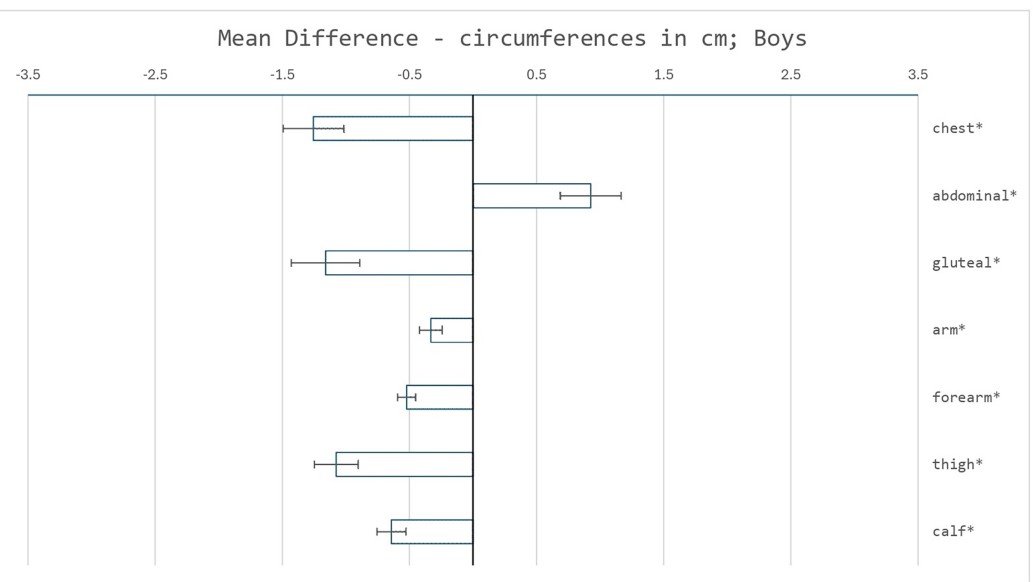

**Figure 3 Mean differences in circumferences measurements (cm) between the reference sample and the recent sample of boys.** Error bars represent the standard error of the mean difference. Asterisks (*) indicate significant differences determined using Welch's test, with $p < 0.05$ adjusted using the Bonferroni correction.

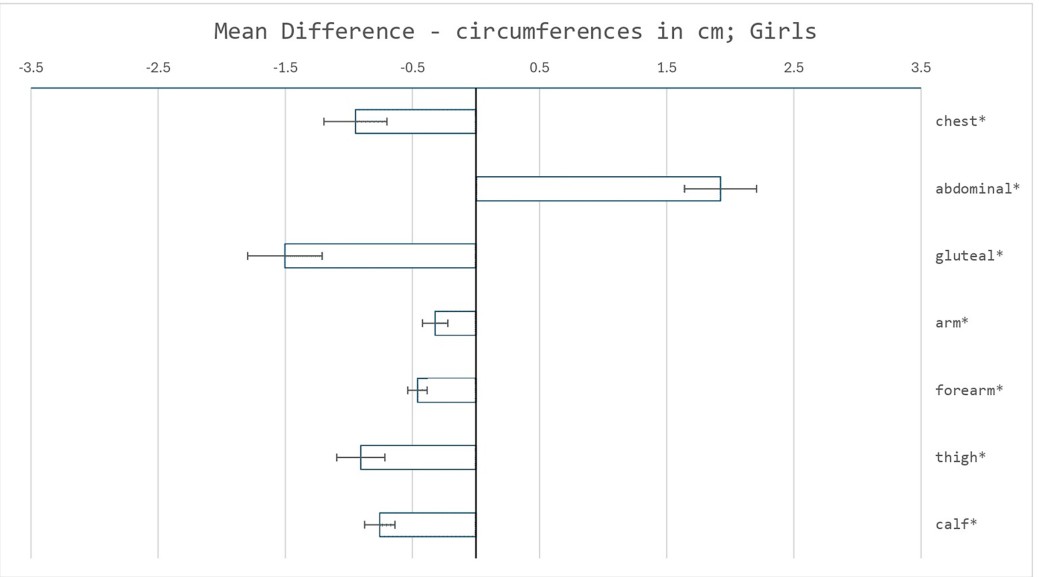

**Figure 4 Mean differences in circumferences measurements (cm) between the reference sample and the recent sample of girls.** Error bars represent the standard error of the mean difference. Asterisks (*) indicate significant differences determined using Welch's test, with $p < 0.05$ adjusted using the Bonferroni correction.

The informational value of the skinfolds and circumferences studied in relation to overall adiposity was analyzed with a correlation analysis to body-fat percentage obtained with BIA. The majority of skinfolds showed moderate correlation, with the highest relevance in the abdominal and subscapular skinfolds. Values were more significant in girls. Tests of the validity of the body-fat percentage estimation showed similar results ($r > 0.7$) when compared to values from BIA (see Table 4).

To evaluate the usability of body fat estimation methods in clinical practice, the bias for each method was calculated using the Bland-Altman method. For the Slaughter method, the bias was found to be 0.759, with the limits of agreement ranging from −6.571 to 8.090. The Durnin method showed a bias of 0.554, with the limits of agreement between −7.376 and 8.484. The graphs for the Bland–Altman method are in the Supplemental Material.

## DISCUSSION

In the evaluation of secular changes in the current sample in comparison to the reference sample, there was a non-significant difference in mean BMI-SDS. On the other hand, a significant increase was found in the sum of four skinfolds, which proves an increase of fat mass. In girls the increase was more substantial, and other methods of fat mass estimation also showed changes. The phenomenon whereby children seemed phenotypically lean, but were over the limit of fat tissue, was described in our pilot sample of preschool children, and termed hidden, or latent obesity (*Sedlak et al., 2017*). In the clinical literature this type of obesity is described as normal-weight obesity in patients with normal values of BMI, but with high metabolic risk and cardiovascular complications (*Maligie et al., 2012*). The cause of latent obesity in childhood can be found in insufficient mechanical stimulation of active

**Table 2 Secular changes in skinfolds and circumferences in boys.**

|  | Statistic | p | Effect size |
|---|---|---|---|
| **Skinfolds** | | | |
| Above hyoid | 2.204 | 0.02797 | 0.1395 |
| Chest I | 3.352 | 0.00087* | 0.2207 |
| Chest II | −2.178 | 0.02992 | −0.1393 |
| Abdomen | 7.617 | <0.00001* | 0.5398 |
| Suprailiac | −0.179 | 0.85817 | −0.0113 |
| Biceps | −1.564 | 0.11852 | 0.099 |
| Forearm | 10.854 | <0.00001* | 0.6946 |
| Subscapular | −3.304 | 0.00101* | 0.1952 |
| Triceps | −0.424 | 0.67171 | −0.0262 |
| Mid-thigh | 2.371 | 0.01816 | 0.151 |
| Patella | −1.497 | 0.13515 | −0.0965 |
| Calf I | −12.587 | <0.00001* | −0.7127 |
| Calf II | −5.918 | <0.00001* | −0.3655 |
| **Circumferences** | | | |
| Chest | −5.267 | <0.00001* | −0.2871 |
| Abdominal | 3.258 | 0.00119* | 0.1915 |
| Gluteal | −4.396 | 0.00001* | −0.2585 |
| Thigh midpoint | −6.258 | <0.00001* | −0.393 |
| Calf | −5.65 | <0.00001* | −0.3601 |
| Arm | −3.728 | 0.00022* | −0.2367 |
| Forearm | −7.4 | <0.00001* | −0.4659 |
| BMI-SD | −0.763 | 0.44562 | −0.0472 |
| Slaughter eq. | 1.5187 | 0.1296 | 0.119 |
| Sum of four, Best | 2.7919 | 0.00546 | 0.180 |

Notes:
* Significance after applying Bonfferoni correction for multiple comparsions.
$H_a$ $\mu_{recent} \neq \mu_{reference}$.

body mass development, especially the skeletal muscles. A sedentary lifestyle preferred from an early age results in underdevelopment of the muscles, especially on the lower limbs, where proportional volume structures are completed with fat tissue. In boys, was proved in this context a highly significant increase in skinfolds above quadriceps while the thigh circumference slightly decreased (*Sedlak et al., 2020*). That results in changes to the ratio of body composition components while overall body weight related to height, *e.g.*, BMI-SDS, shows normal results.

Studies have shown changes in the distribution of fat mass with age. While children under 5, or 7 have more fat mass on the limbs compared to the trunk, gradually fat tissue increases on the trunk to a relatively balanced ratio (*Hajniš, Pařízková & Petrásek, 2003*). Only in the pubertal period does the distribution of subcutaneous mass start to show intersexual differentiation, in girls with a predilection into the glutes and thighs (centrifugal type), in boys with a predilection to the trunk and abdominal area (centripetal

**Table 3 Secular changes in skinfolds and circumferences in girls.**

|  | Welch's $t$ | $p$ | Effect size |
|---|---|---|---|
| **Skinfolds** | | | |
| Above hyoid | 3.219 | 0.00138* | 0.2095 |
| Chest I | 2.992 | 0.00292 | 0.1979 |
| Chest II | −1.448 | 0.14834 | −0.0951 |
| Abdomen | 9.351 | <0.00001* | 0.6591 |
| Suprailiac | 1.447 | 0.14857 | 0.0978 |
| Biceps | −0.309 | 0.7572 | −0.0199 |
| Forearm | 10.873 | <0.00001* | 0.7015 |
| Subscapular | −1.961 | 0.05037 | −0.1225 |
| Triceps | −0.728 | 0.46722 | −0.0465 |
| Mid- thigh | 1.191 | 0.23433 | 0.0742 |
| Patella | −3.342 | 0.00089* | −0.2024 |
| Calf I | −9.351 | <0.00001* | −0.582 |
| Calf II | −3.558 | 0.00041* | −0.221 |
| **Circumferences** | | | |
| Chest | −3.814 | 0.00015* | −0.2284 |
| Abdominal | 6.798 | <0.00001* | 0.4508 |
| Gluteal | −5.125 | <0.00001* | −0.33 |
| Thigh | −4.791 | <0.00001* | −0.3034 |
| Calf | −6.368 | <0.00001* | −0.4077 |
| Arm | −3.241 | 0.00128* | −0.2125 |
| Forearm | −5.951 | <0.00001* | −0.3929 |
| BMI-SD | 1.209 | 0.2273 | 0.0763 |
| Slaughter eq. | 9.353 | <0.00001* | 0.7763 |
| SUM4_Best | 3.805 | 0.00016 | 0.2558 |

**Notes:**
* Indicates significance after applying Bonfferoni correcetion for multiple comparsions.
$H_a$ μ recent ≠ μ reference.

type) (*Moreno et al., 1997*, *1998*; *Hajniš, Pařízková & Petrásek, 2003*). In the recent children we can find differences from an early age; even in preschool children, there is a significantly higher cumulation of subcutaneous fat tissue on the trunk, with a predilection to the abdominal area. The increase in skinfold on the abdomen was highly significant in comparison to the reference sample, with high clinical relevance, in both sexes. The next region was on the thigh, where skinfold thickness has not changed significantly, but there has been a significant decrease in circumference. The higher contribution of fat mass proves insufficient development of muscle mass in this region, which can be contextualized with the impact of sedentary behavior in child populations mentioned above. The central distribution of fat mass as a phenomenon of the current development of the child and adolescent population has also been found in a range of international studies evaluating secular changes in adiposity and the distribution of subcutaneous fat (*Nagel et al., 2009*;

**Table 4 Association between selected anthropometric parameters and body fat (%) estimated by BIA in recent sample.**

|  |  | M | F |
|---|---|---|---|
| Abdomen SF | Pearson's $r$ | 0.514 | 0.687 |
|  | $p$-value | <0.00001 | <0.00001 |
| Suprailiac SF | Pearson's $r$ | 0.443 | 0.601 |
|  | $p$-value | <0.00001 | <0.00001 |
| Subscapular SF | Pearson's $r$ | 0.533 | 0.669 |
|  | $p$-value | <0.00001 | <0.00001 |
| Forearm SF | Pearson's $r$ | 0.496 | 0.576 |
|  | $p$-value | <0.00001 | <0.00001 |
| Abdominal C | Pearson's $r$ | 0.329 | 0.573 |
|  | $p$-value | <0.00001 | <0.00001 |
| Forearm C | Pearson's $r$ | 0.193 | 0.452 |
|  | $p$-value | 0.00094 | <0.00001 |
| Slaughter eq. | Pearson's $r$ | 0.636 | 0.75 |
|  | $p$-value | <0.00001 | <0.00001 |
| Deurenberg eq. | Pearson's $r$ | 0.659 | 0.745 |
|  | $p$-value | <0.00001 | <0.00001 |
| Durnin[a] eq | Pearson's $r$ | 0.647 | 0.785 |
|  | $p$-value | <0.00001 | <0.00001 |
| Sum of four, Best | Pearson's $r$ | 0.605 | 0.731 |
|  | $p$-value | <0.00001 | <0.00001 |
| BMI-SDS | Pearson's $r$ | 0.597 | 0.731 |
|  | $p$-value | <0.00001 | <0.00001 |

**Notes:**
[a] Biceps skinfold was measured using Best caliper and transformed using population specific transformation table.
M, boys; F, girls; SF, skinfold; C, circumference.

*Leal et al., 2015*; *Suder, Gomula & Koziel, 2017*; *De Santis Filgueiras et al., 2019*; *Żegleń et al., 2022*).

Worries about the validity of a clinical methodology that works with skinfold thickness have risen alongside changes in the distribution of fat mass. One of the standard skinfolds, above the triceps (and coincidentally above the biceps as well) is very stable in a secular context, and as revealed by our study, its thickness in preschool children remains the same after 35 years. The same trends are also shown in other studies for the prepubertal and pubertal categories (*Nagel et al., 2009*; *Leal et al., 2015*). Due to the trend of the prevalence of truncal fat distribution, it is more informative to prefer measurements of skinfolds on the abdomen and chest. The highest increment was observed in the abdominal skinfold, and it would therefore be advisable to use this skinfold as an obesity marker in clinical use, as a valid and sensitive evaluation of the reduction process.

In clinical practice, measurements of circumferences of various parts of the body are also used to evaluate the state of nutrition. Of the various circumferences, the most valid marker of adiposity is waist circumference, which needs to be standardized with the use of the WHtR index in children due to changes with growth. This index highly correlates to

the amount of visceral fat, and is consequently a beneficial criterion for the evaluation of metabolic risk (*Sardinha et al., 2016*; *Lo et al., 2016*). Its use is valuable for validating weight reduction regimes, where BMI-SDS values need not show a substantial decrease, but waist circumference reduction shows a decrease in visceral fat and the health risk of obesity. For the evaluation of active body mass, *e.g.*, skeletal muscle mass, in children, the best marker is reported to be the midpoint circumference of the thigh with the skinfold in this region. Their ratio can reveal latent obesity. Inadequate muscle growth in recent children is also documented by a significant decrease in the maximal circumference of the forearm with a clinically relevant increase of the skinfold.

As calculated results of bias in our study revealed, in the evaluation of fat mass in children with the possibility of simple calculations from skinfolds such as the most used, the Slaughter equation (*Żegleń et al., 2022*), and even the less commonly used the Durnin and Deurenberg equations (*Rudnev et al., 2020*; *Penagini et al., 2021*; *Soylu et al., 2021*), sufficient diagnostic ability in children was proven in relation to adiposity and BMI-SDS. However, validity is limited in individuals with extreme height. In children with short stature results are underestimated, while by contrast in tall statures overestimation of real status occurs, which can result in incorrect weight categorization.

The possibility of using anthropometric characteristics extends to direct evaluation of health risk, such as direct evaluation of the risk of cardiovascular diseases with the waist to height ratio (*Lewitt & Baker, 2020*), of cardiometabolic risk with the use of waist circumference and skinfolds (*de Quadros et al., 2019*), and of metabolic syndrome with waist circumference (*Muhanna et al., 2022*), alongside prediction of hypertension with the use of abdominal skinfold thickness (*Wang et al., 2022*).

A limitation of the study is the unequal sample sizes of the recent and reference samples. Furthermore, we acknowledge the possibility of selection bias due to measuring children only from urban areas, and the voluntary participation based on the will of children's legal representatives in the recent sample. We recognize that some parents decide the participation of their children based on the possible consequent results, and therefore may opt not to participate because of worries about negative results in terms of obesity.

## CONCLUSIONS

Alongside the trend of the increasing prevalence of excess weight and obesity, secular changes in the distribution of fat mass have been documented. From an originally balanced distribution, a shift has begun towards truncal cumulation. This is supported by changes in skinfold thicknesses and body circumferences that are part of clinical evaluation in children with obesity. The aim of this study was to analyze secular changes in 13 skinfolds and eight circumferences in a population of preschool children, and to evaluate the validity of methods based on these parameters in a clinical environment.

In most of the skinfolds thickness did not change. Exceptions were a significant increase in skinfolds on the abdomen, chest I and forearm, with high clinical relevance in both sexes. In circumferences only decreases were observed, except for the abdominal circumference, which showed a significant increase. This illustrates the trend of a decline of circumferences with the rise of skinfold thicknesses in corresponding regions, displaying

an escalation of adiposity with a decrease of muscle mass. This trend has been termed "hidden", or latent, obesity, that is defined as normal BMI-SDS and the overlimit adiposity of a child. Even though our study focused on a rather narrow age group, our findings are applicable more widely, as the issue of obesity and excess weight is spread across child population.

Our findings support the need for mindful use of methods evaluating body composition in children, and for acknowledging secular changes in the evaluation of body composition. Therefore, we propose the use of the abdominal skinfold and evaluation of muscle mass through use of circumferences with corresponding skinfolds. The most valid regions are the midpoint thigh and the maximal circumference of the forearm. These findings offer significant importance and value for practical applications.

## ACKNOWLEDGEMENTS

We would like to thank all the participants, their guardians and the schools that agreed to cooperate to accomplish our study. Alastair Millar provided a linguistic review.

### Funding

This work was supported by the Charles University Grant Agency (GA UK No. 206823). The funders had no role in study design, data collection and analysis, decision to publish, or preparation of the manuscript.

### Grant Disclosures

The following grant information was disclosed by the authors:
Charles University Grant Agency: GA UK No. 206823.

### Competing Interests

The authors declare that they have no competing interests.

### Author Contributions

- Anna Vážná conceived and designed the experiments, performed the experiments, authored or reviewed drafts of the article, and approved the final draft.
- Jan M. Novák conceived and designed the experiments, analyzed the data, prepared figures and/or tables, authored or reviewed drafts of the article, and approved the final draft.
- Robert Daniš conceived and designed the experiments, authored or reviewed drafts of the article, and approved the final draft.
- Petr Sedlak conceived and designed the experiments, authored or reviewed drafts of the article, and approved the final draft.

### Human Ethics

The following information was supplied relating to ethical approvals (*i.e.*, approving body and any reference numbers): IRB approval number 2017/23.

The Institutional Review Board of the Faculty of Science, Charles University in Prague. approved this study.

## Data Availability

The raw data is available in the Supplemental Files.

## Supplemental Information

Supplemental information for this article can be found online at http://dx.doi.org/10.7717/peerj.18695#supplemental-information.

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
