# Peer review of "Adiposity and body fat distribution based on skinfold thicknesses and body circumferences in Czech preschool children, secular changes"

_PeerJ, doi:10.7717/peerj.18695_

## Round 0.1 · original submission · Minor Revisions

This manuscript presents an interesting investigation into secular changes in skinfold thickness and body circumferences in preschool children. While the study addresses an important topic with potential implications for clinical practice and public health, several areas require revision to enhance clarity, precision, and overall impact.
General Comments:
The manuscript demonstrates potential but needs significant improvements in structure, clarity, and adherence to academic writing conventions. The following feedback focuses on specific areas requiring attention:
1. Paragraph Structure and Flow:
Many paragraphs lack a clear focus and logical flow, hindering comprehension. Sentences within paragraphs often seem disjointed, lacking smooth transitions and a cohesive narrative. Consider revising with a clear introductory sentence that sets the theme, followed by supporting sentences that delve deeper, and a concluding sentence that summarizes the key takeaway. Consider working with an editor experienced in academic writing to refine the manuscript's flow and clarity.
Example:
Take, for example, the first paragraph in the discussion section discussing body composition and latent obesity to exemplify this issue. The paragraph jumps right into results without first establishing the central theme or purpose. It shifts between different ideas (differences in height/BMI, increase in skinfolds, latent obesity, causes of latent obesity, and consequences) without smooth transitions. The information about "hidden obesity" is introduced abruptly, making it hard to follow the logic. The paragraph ends with details about motor abilities without tying it back to the main point or leading to the next idea. The connection between the findings on skinfold thickness and the concept of latent obesity is not immediately apparent.
2. Introduction and Secular Changes:
The introduction lacks a comprehensive overview of the existing literature on secular changes in childhood growth and body composition. While some relevant studies are mentioned in the discussion, they should be integrated into the introduction to provide context and rationale for the current study. Please revise the introduction to provide a more robust overview of the relevant literature, highlighting the knowledge gap and the contribution of this study. Elaborate on the theoretical framework of secular changes and how it applies to your research question.
3. Presentation of Data and Results:
The presentation of data and results could be improved to adhere to academic conventions. The term "clinical significance" is used without a clear definition, and the results lack specific numerical data. Table 1 provides limited information and could be moved to supplementary materials. Revise the results section to ensure clear and concise data reporting, including precise descriptions of effect sizes and statistical significance. Instead of stating "clinically significant," specify the criteria used to determine significance, such as effect size thresholds or confidence intervals. When reporting Cohen's d, provide the exact value and interpretation (e.g., "d = 0.60, indicating a medium effect size"). Consider using graphs or figures to present key findings more effectively, such as the Bland-Altman method. For example: "The Bland-Altman plot revealed a small but statistically significant bias, with Method A measuring slightly higher than Method B on average. However, the limits of agreement were relatively narrow, suggesting that the two methods generally agree well. No clear trends or outliers were observed in the plot."

4. Methodology and Sample:
The methodology section requires more detail about the current and previous sample recruitment procedures. Clarify how the samples were selected and whether they represent the population of interest. Provide specific details about the recruitment process: How were participants contacted? What was the response rate? Discuss any potential sampling biases and how they might affect the generalizability of the findings.
5. Abstract:
The abstract requires significant revision to reflect the study's key findings and improve clarity accurately. The introduction is overly broad, the results lack specific data, and the conclusion overreaches the scope of the study. Streamline the introduction to focus on the knowledge gap related to secular changes in skinfolds and circumferences. Include key numerical results and effect sizes. Refine the conclusion to summarize the key findings and their implications for clinical and research applications. Define the criteria for assessing clinical relevance and provide more details about the methodology and statistical analysis.

6. People-First Language:
The language used in the manuscript could be more sensitive and person-centered. Adopt a people-first language to emphasize the individuals behind the conditions or characteristics being studied.

For example, instead of "the long-standing worldwide problem that is obesity," consider phrasing like "the widespread prevalence of obesity" or "the global challenge of obesity." This subtle shift in language promotes respect and avoids potential stigmatization.
Another example is instead of saying "obese adults" or "obese children," we now say "adults with obesity" and "children with obesity." This shift in wording emphasizes the person first and their condition second.
For example, the sentence “almost three quarters of adults with obesity were not obese in childhood but just over half of obese children remains obese in adolescence and 80% of obese adolescents are obese in adulthood” could be replaced by “While most adults with obesity had a healthy weight as children, childhood obesity can persist. Over half of children with obesity remain at an unhealthy weight in their teens, and 80% of those teens go on to have obesity in adulthood.

7. Title:
The current title accurately reflects the study's correlational aspect, highlighting the relationship between skinfold measurements and fat composition. However, it doesn't explicitly mention the crucial element of secular changes, which is the research's foundation.

Overall
This study has the potential to make a valuable contribution to the understanding of secular changes in body composition. However, the manuscript requires substantial revisions to address the concerns outlined above. By focusing on clarity, precision, and adherence to academic writing conventions, the authors can significantly enhance the impact and readability of their work. The authors should clearly and explicitly highlight the key results that will impact clinical practice.

Reviewer 1 ·

Basic reporting

1) The tittle is interesting, nonetheless the phrase ‘development’ may not be relevant to the discussion of the study. I suggest that phrase ‘change’ or ‘secular change’ is more appropriate.

2) The first paragraph of introduction is too long. Please consider reorganizing the introduction for greater clarity. Authors might improve their writing by giving the significance and usefulness of the study.

Experimental design

1) Please specify if the study is a cross-sectional or cohort design. The Abstract mentions cohort studies, yet the Methods and Sample sections mention cross-sectional research. What are the data resources for the reference cohort (from 1990) or the data collected (or reported) in 1990? Please explain how cohort data are compared to cross-sectional data to determine mean differences.

2) The figures are relevant, well-labelled, and detailed. However, the quality might be improved by raising or thickening the type labels for easier readability.

3) What do you mean by significant increase in females (p≥0.01) - Line 220 and (p=1) - Line 221? Please refer to Tables 2 and 3.

4) Please specify whether the samples utilized in Table 4 are recent, reference, or both.

Validity of the findings

1) The author should clarify that examining children of all ages (4 to 6.9 years) rather than stratifying by age is suitable for understanding growth differences.

Additional comments

The manuscript is clearly written, unambiguously, and in professional English, however a few improvements can be made. The research questions are clearly defined and relevant. The method is explained in sufficient detailed and information to enable replication. The data and analysis are statistically valid and controlled. Conclusions are well expressed.

·

Basic reporting

This paper is interesting and important regarding the Czech publich health policy. However, there are some issues that should be clarified.

Experimental design

Clear

Validity of the findings

clearly explnained

Additional comments

SeeL. 88: Active or muscle mass? Active or lean mass?
L. 94: Should be: "A study" instead of "This study".
Introduction: It is important to better clarify the research gap. Why is this important? Because you have changed during time. So, there's a need to standarize the procedures.

Methods: There's nothing regarding the sample and statistical power.

Discussion: Limitations should be here.

Provide practical aplications before conclusion.

---

## Round 0.2 · Minor Revisions

Dear Authors,

Thank you for your thorough efforts in addressing the editor's and reviewers’ comments. I appreciate the modifications made to enhance the manuscript and recognize the work put into responding to the feedback provided. Below, I have outlined a few additional suggestions to further improve the clarity and readability of your article.

Title Recommendation
To enhance clarity, I suggest revising the title to:
“Secular Trends in Body Fat Distribution and Adiposity Among Czech Preschool Children Using Skinfold and Circumference Measures.”


Language and Readability
While the revisions have improved certain sections, the overall readability of the manuscript remains challenging. To ensure clarity and improve the flow, I strongly recommend that you work with a professional English editor who can assist in refining the language throughout the manuscript.

Thank you again for your continued efforts. I look forward to seeing the improvements in the final version.

---

## Round 0.3 · accepted · Accept

I have carefully reviewed the revised manuscript and am satisfied that the authors have adequately addressed the previous concerns. This manuscript is now ready for publication.

·

Basic reporting

The manuscript is now corrected.

Experimental design

ok

Validity of the findings

ok

Additional comments

The authors adressed my concerns.